# A New High-Throughput Focused MeV Ion-Beam Analysis Setup

**Sören Möller** [1],*, **Daniel Höschen** [1], **Sina Kurth** [1], **Gerwin Esser** [1], **Albert Hiller** [1], **Christian Scholtysik** [2], **Christian Dellen** [1] and **Christian Linsmeier** [1]

1   Institut für Energie- und Klimaforschung, Forschungszentrum Jülich GmbH, 52425 Jülich, Germany; d.hoeschen@fz-juelich.de (D.H.); sina-kurth@gmx.de (S.K.); g.esser@fz-juelich.de (G.E.); a.hiller@fz-juelich.de (A.H.); c.dellen@fz-juelich.de (C.D.); ch.linsmeier@fz-juelich.de (C.L.)
2   Peter Grünberg Institut–9, Forschungszentrum Jülich GmbH, 52425 Jülich, Germany; c.scholtysik@fz-juelich.de
*   Correspondence: s.moeller@fz-juelich.de

**Abstract:** The analysis of material composition by ion-beam analysis (IBA) is becoming a standard method, similar to electron microscopy. A pool of IBA methods exists, from which the combination of particle-induced-X-ray emission (PIXE), particle induced gamma-ray analysis (PIGE), nuclear-reaction-analysis (NRA), and Rutherford-backscattering-spectrometry (RBS) provides the most complete analysis over the whole periodic table in a single measurement. Yet, for a highly resolved and accurate IBA analysis, a sophisticated technical setup is required integrating the detectors, beam optics, and sample arrangement. A new end-station developed and installed in Forschungszentrum Jülich provides these capabilities in combination with high sample throughput and result accuracy. Mechanical tolerances limit the device accuracy to 3% for RBS. Continuous pumping enables $5 \times 10^{-8}$ mbar base pressure with vibration amplitudes < 0.1 μm. The beam optics achieves a demagnification of 24–34, suitable for μ-beam analysis. An in-vacuum manipulator enables scanning $50 \times 50$ mm$^2$ sample areas with 10 nm accuracy. The setup features the above-mentioned IBA detectors, enabling a broad range of analysis applications such as the operando analysis of batteries or the post-mortem analysis of plasma-exposed samples with up to 3000 discrete points per day. Custom apertures and energy resolutions down to 11 keV enable separation of Fe and Cr in RBS. This work presents the technical solutions together with the quantification of these challenges and their success in the form of a technical reference.

**Keywords:** ion-beam analysis; Rutherford-backscattering spectrometry; nuclear reaction analysis; particle induced x-ray spectroscopy; material analysis

## 1. Introduction

Ion beam analysis (IBA) is a versatile group of methods using a broad range of projectiles with kinetic energies per ion in the order of 1 to 10 MeV and ion currents in the range of pA to μA to probe surface near ~10 μm concentrations of a sample composition. The method is, in most cases, non-destructive and reference free. It yields depth profiles of element and isotope amounts and concentrations when combined with computer based interpretation. Furthermore, a combination with scanning methods enables μm resolved compositional tomography when using μm sized ion-beam diameters (so-called "microbeam" variants). For these reasons, the applications of IBA span from historical paintings over thin-film technology up to material development, with each application having its special requirements on the analysis setups.

Usually, the measurements are conducted in vacuum due to the detrimental effects of matter on ion beams. The measurements rely on the detection of the products of the ion-matter interaction. These can be secondary electrons, gamma rays, x-rays, recoils, scattered projectiles, and nuclear reaction products. Many different techniques are available to exploit the information hidden in these products [1]. The combination of the detection of

several products in parallel proved to be useful for determination of unambiguous results, since for example, backscattering from a heavy element present deep in the sample or from a light element present at the sample surface may result in the same spectra [2]. A combination with a particularly low amount of ambiguity and a straightforward integration is nuclear-reaction-analysis (NRA), Rutherford-backscattering-spectrometry (RBS), and particle-induced-X-ray emission (PIXE), also referred to as Total-IBA [3]. This combination can provide ppm detection limits and sub-% accuracy for light (NRA), intermediate (PIXE), and heavy (PIXE+RBS) elements in a self-consistent way, which are particular advantages compared to electron beam based analysis. These advantages in combination with the isotopic sensitivity of RBS, NRA, and PIGE are particularly interesting for materials research, for example for nuclear fusion reactors [4–6].

For this set of methods, the ion-beam and the end-station properties define the analysis properties. A high intensity ion beam with a small beam diameter is required to obtain decent counting statistics (=result accuracy) and lateral spatial resolution (=beam spot size). This can be achieved with beam optics that flexibly focus the beam onto the samples with the minimal spot sizes depending on the initial beam emittance from the accelerator and the properties of the beam optics. Numerous devices exist, regularly achieving spot sizes in the order of 1 μm by combining focusing with modern ion sources and accelerators, leading to high-brightness ion-beams [7,8].

This reference work introduces a new setup combining a magnetic quadrupole focusing system with a custom vacuum chamber and sample positioning system, which potentially realizes μm spot sizes with a special emphasis on high measurement throughput. The technical issues arising from sample positioning on the μm scale in a non-magnetic vacuum environment with high sample throughput is discussed and solutions are presented. Unfortunately, the available 1.7 MV Cockcroft–Walton tandem accelerator with a Duoplasmatron source (both dating 1983) limited the achievable spot-sizes of the presented device due to the 1–2 orders of magnitude larger energy-normalized emittance (>20 $\pi$ mm mrad (MeV)$^{1/2}$) compared to modern systems. This limits the minimal spot-sizes correspondingly.

## 2. Materials and Methods

The new setup combines a quadrupole triplet focusing magnet, an UHV chamber, a vibration isolated table, and the detector setup connected to a beam-line and a tandem ion accelerator, as shown in Figure 1. Generally, the relative alignment accuracy between a setup components and the sample limit the base accuracy and result reproducibility of an IBA setup. The counting performance improves with smaller dimensions and distances, since e.g., a smaller sample to detector distance increases the detector solid angle at a given detector size. This induces technical layout conflicts, since smaller dimensions limit sample dimensions and increase relative alignment inaccuracy with a given manufacturing tolerance. A technical setup providing precision in the percent range in combination with the possibility of frequent sample exchange, high sample throughput, and the requirements on precise angular alignment and low ion doses required the development of a new analysis setup. This new setup is called μNRA and will be described in the following text.

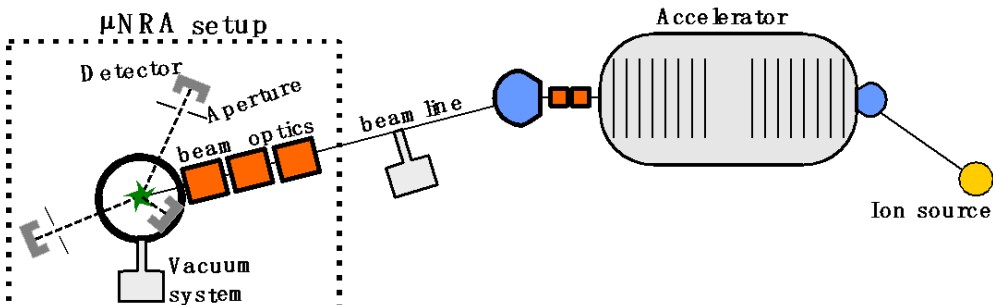

**Figure 1.** Sketch of the μNRA setup with the required modules and the target (green star) attached to a tandem accelerator. Blue objects depict dipole magnets. Orange squares depict ion beam focusing devices.

## 3. Vacuum Chamber and Beamline

Figure 2 shows the design of chamber and its supporting systems. The manufacturing results in 0.1 mm tolerances at the points where an impact on measurement angles and distances is relevant, e.g., the ion-beam impact angle. The beamline is made from CF100 316 stainless steel vacuum parts and has a length from sample to switching magnet of about 5 m. The beamline is equipped with the identical turbo- and fore-pump as the µNRA chamber and reaches a base pressure of $2 \times 10^{-9}$ mbar. Two Faraday cups and one beam profile monitor are installed for beam diagnostics, which can be moved out of the beam path during the sample analysis. An iron-core steering magnet powered by two remote-controlled power supplies (0.2 mA steps) after the first (object) aperture provides additional flexibility for the beam control. A second aperture is installed right before the focusing magnets (see Figure 2). Together these apertures collimate the beam in terms of dimension and divergence. Both apertures feature a fixed 5 mm diameter hole followed by four motor controlled blocks representing the four sides of the adjustable rectangular aperture. All parts are equipped with tantalum shields for radiation levels <10 µSv/h in the laboratory when operating with H, D, and He ions. Heavier ions cannot be accelerated to relevant energies.

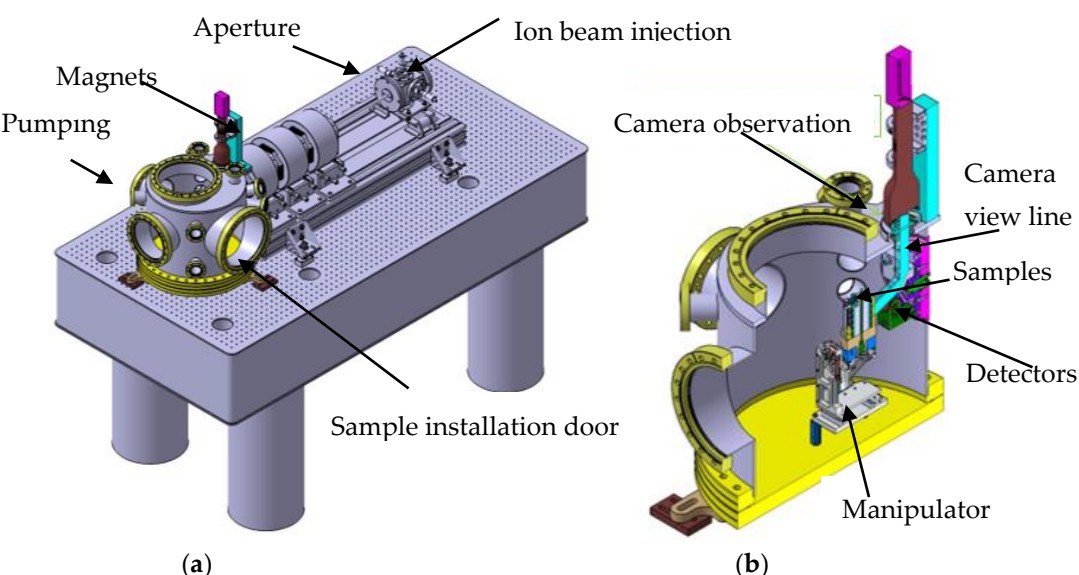

(**a**)  (**b**)

**Figure 2.** CAD drawings of the µNRA setup (dotted square in Figure 1). (**a**) The µNRA vacuum chamber (flanges in yellow) together with the three focusing magnets and an aperture on a vibration-damping table. (**b**) Detailed crosscut of the vacuum chamber with sample manipulator, observation, charged particle detectors, and holder.

The sample chamber is manufactured from 316 stainless steel and mechanically polished from the inside. It features 18 CF flanges for feedthroughs, detectors, sample observation, and sample access together with an individually designed short flange for flexible connection of the beam tube going through the focusing magnets to the analysis chamber. Two self-assembled thermocouples of type K with 500 V isolation voltage and 1.5 K accuracy, electrical wirings for in-vacuum connections for example of lithium batteries as depicted in Figure 3, and a multi-pin feedthrough equipped with 0.2 mm Kapton isolated wires are installed here. The multi-purpose wiring enables for example charging and discharging of batteries during IBA measurement via an external cell tester.

During the assembly of the beamline and chamber, special attention is paid on avoiding stray electro-magnetic fields in the beamline and the detector region in order to provide compact ion beams and reproducible scattering angles. Magnetic field measurements revealed the vacuum gauges (low stray field, Pfeiffer Vacuum PKR 360), the (electro-pneumatic) valve actuators, and the magnetic-bearing turbo-molecular pumps as critical sources of stray fields with strength in the mT range. To mitigate the stray field influence

on the ion beam and charged products, these devices are mounted on extension flanges providing sufficient distance for the field amplitude to reduce below 10 µT at the beamline center and in the measurement chamber, respectively. The fields are measured using a hand-held magnetic compass with an effective handling resolution of 1 µT (0.01 µT digital resolution). The field dropped below the detection limit at a distance of 450 mm for the turbo-molecular pump and 200 mm for the vacuum gauges and actuators, respectively.

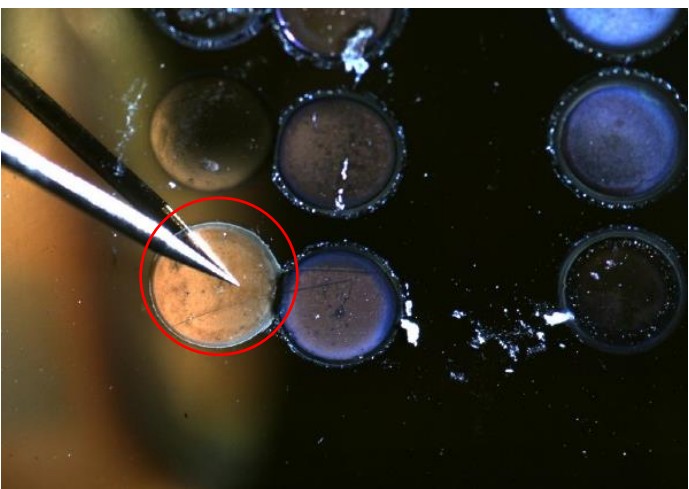

**Figure 3.** Camera sample image of an in-situ contact of an all-solid state lithium thin-film battery (red circle) using a W contact needle (silver part on the left) attached to an isolated 2 pole DC feedthrough (not shown) with an outside cell tester. The cell to source wiring accounts to 1 Ω DC resistance.

The beamline is mounted and aligned with the help of a leveling camera. All flanges are centered to $\pm 1$ mm (camera accuracy) around the ideal track, including beamline, magnets, apertures, and the sample chamber on its table (Figure 2). After centering, the chamber is further corrected for rotational alignment with the help of adjusting screws and a laser beam injected 5 m away at the backside of the switching magnet to a precision of $10^{-4}$ degree against the plane of vertical axis and beam and furthermore against the perpendicular plane by a level of 0.01° precision rating. After the beamline and chamber assembly, the sample manipulator is installed into the chamber. The degree of precision achieved in mounting allows for fine adjusting all parts with their individual fine adjusting screws in the next stage.

Thermal drifts of the structures relevant for relative beam spot movements between the accelerator, apertures, and sample are assessed by ambient measurements over one week in summer. The room temperature variations are found to be in the order of up to 1 K/h over an operational day. The steepest gradient typically observed lies between 10:00 to 12:00. The overall temperature variations between 8:00 and 20:00, the usual working hours, stayed within 2 K. The maximum assumed temperature difference is 1 K, as all metal components provide decent thermal conductivity for equilibration. The measurement also revealed changes of relative humidity in the range of 30–70%. To avoid humidity related swelling, only metals were used in the whole support structures.

For ion beam measurements requiring long acquisition times, a relative positional drift between last aperture and sample (which are both on the same table) is considered as the relevant quantity, since the ion beam focusing projects the aperture opening onto the analyzed spot on the sample. Distance changes are not relevant, as they will be negligible compared to the focal length discussed in the following sections. The main concern are vertical displacements induced by thermal expansion of the support structures. All materials below the table surface are selected to be identical. The part above the table surface is assumed to induce the same displacement to all parts mounted on it with a relevant height of the beamline above the table of 150 mm. With the thermal expansion coefficient of aluminum ($2.3 \times 10^{-5}$/K) used for construction X-profiles minus the one of

stainless steel 316 ($1.6 \times 10^{-5}$/K), a maximum displacement of 1 μm is calculated as the upper limit neglecting temperature conduction.

## 4. Pumping System and Vibrations

The pumping system of the vacuum chamber is constructed to provide a low base pressure ($5 \pm 2 \times 10^{-8}$ mbar) for stable sample surfaces and ion energy losses <0.1 keV together with quick sample exchange and pump-down times in the order of 10 minutes. In addition, the pumping system has to avoid inducing magnetic stray fields and vibrations into the measurement chamber. The reduction of these quantities has an important role for positioning accuracy, charged particle analysis, and the measurement of small electrical signals in the detector system (vibration-induced voltages). Therefore, the vacuum system consists of three parts, subdivided by gate valves and elastic elements. These parts are the fore-vacuum pump, the turbo-pump, and the μNRA chamber. Every part is equipped with a vacuum gauge and a venting valve for safe access and easy maintainability. The pumping system is installed at a CF100 side port of the sample chamber. An edge-welded bellow separates beam-line and μNRA chamber.

The rotary vane fore-pump is found primarily responsible for the vibrations. The turbo-molecular pump (Pfeiffer 300M) works on fully contactless magnetic bearings, strongly mitigating its external vibration amplitude. Rigid vacuum lines can transmit those vibrations to the samples and devices in the analysis chamber. For reducing this transmission, low stiffness vacuum components are installed with the goal of reducing the vibration amplitude well below 1 μm. To quantify the success of this approach, the vibrations amplitude is measured by a high-accuracy LK-H052 laser distance meter from KEYENCE. The displacement sensor works with a repeatability of 0.025 μm, a spot diameter of 50 μm, and a measurement-frequency of 50 kHz (maximum detectable vibration frequency of 25 kHz), enough to cover the first few harmonics of all possible mechanical frequencies. The laser is located on a stand next to the μNRA table. Due to the impact of the artificial lightning on the measurement, the measurements are conducted with lights off. The laser is positioned on the different analysis points. For each point, the distance is recorded for 60 s, yielding 3 million points. In steady-state, electronic noise and vibrations are assumed to feature different spectral distributions with vibrations following certain rotational frequencies such as 50 Hz. This assumption allows neglecting the white noise in the Fourier spectrum. For the evaluation, the data are corrected for a baseline-shift and the vibration amplitude is taken as the 3σ (99.73%) value of the real spectrum.

First, vibrations of the concrete floor and the supports are excluded by Laser measurements on several static objects and walls. All measurements resulted in signals at the electronic noise limit. The amplitude of oscillation of the vacuum fore-pump in operation is measured to 45 μm for a 1-phase and 30 μm for a 3-phase motor of the same pump type. With its significantly lower vibration levels, the 3-phase pump is selected for all further measurements. The turbo-pump is selected as second measurement point, since it features a vibration amplitude above the detection limits. Fitting different components between fore-pump and turbo-pump enables further optimization of the vibration amplitude. Table 1 lists the measurement results. Softer flexible components reduce the vibration transmission to a level of 1.6 μm, a factor 20 below the source level (fore-pump). The vibration amplitude at the samples will be significantly smaller, since another flexible element and the rigid support table of the vacuum chamber further mitigate the vibrations. At the sample position, no signal is detectable anymore in this situation equivalent to a vibration amplitude <0.1 μm.

**Table 1.** Amplitude of oscillations at the turbo-pump with different damping components in between turbo-pump and fore-pump. Plastic clamps and edge-welded bellows reduce the transmission of the vibration to an acceptable level. The base amplitude of the fore-pump is 30 μm. The electronic noise background is 0.1 μm. Plastic connections reduce the vibration transmission and allow for a galvanic separation.

| Connecting Tubing and Clamping | | | Vibration Amplitude [μm] |
|---|---|---|---|
| Al KF clamp | Flexible hose | - | 3.18 |
| Plastic KF clamp | Flexible hose | - | 3.03 |
| Al KF clamp | Flex. hose + Edge welded bellow | Al clamp | 2.06 |
| Plastic KF clamp | Flex. hose + Edge welded bellow | Plastic clamp | 1.63 |
| Al KF clamp | Flex. hose + Edge welded bellow | Plastic clamp | 1.60 |

## 5. Optical Observation of Samples

A FLEA3-FL3-U3-88S2C color camera with a 8.8 MPixel 1/2.5″ detector is combined with a 0.28× tele-centric lens from Techspec with a working distance of 180 mm, providing a field of view of about 20 mm, a depth of sharpness of ±7 mm, and a tele-centricity of <0.03° for sample observation. The camera observes the samples through a vacuum window and a planar mirror positioned above the ion-beam tube in the μNRA chamber, see camera view line in Figure 2b. The mirror position above the ion beam axis requires a rotation of the mirror normal of 30° against the sample normal (instead of the ideal 45°). Due to the camera tele-centricity preserving all distances over the whole field of view, even in a rotated situation and for an offset between ion-beam and camera observation spot, samples are always observed virtually head-on. This eases beam position and size/area determination on scintillators exposed to the beam, but results in a thin sharpness line due to the mirror angle. No error will be connected to storing the beam outline on the camera image and navigating on non-scintillating samples. A position verification by the means of the camera system is required after each sample exchange, as mechanical tolerances and changes in the beam optics limit the accuracy of the relative positioning of samples and ion beam. A far-field microscope could have offered a better spatial resolution than the tele-centric lens, but the chosen solution provides the larger field of view, higher depth of field, tele-centricity, and also significantly lower costs. Imaging by beam scanning is also possible, but slow compared to the optical alignment. Upon installation, adjusting screws allow tilt, rotational, and distance adjustments by observing samples on the adjusted manipulator, see Section 7 for further details.

The tele-centric camera setup is first calibrated in a test stand with the help of a standardized vacuum compatible test pattern also used for the later beam resolution tests (Figure 4). The best spatial resolution was found by varying the angle of incidence, lens aperture, the test patterns location in the camera field of view, the test patterns lighting, the lens aperture opening, and the distance between lens front and test pattern. The distance between two line centers on the test pattern compared to the number of pixels between two lines, yields the pixel width calibration, as shown in Figure 4. Ten different groups are investigated with variations from 5.521 μm/pixel to 5.861 μm/pixel. On average, a pixel corresponds to 5.71 ± 0.12 μm, compared to nominal calibration of 5.5 μm/Pixel according to the datasheets. A change of the test patterns location in the camera field of view had no influence on this calibration, proving the lens' tele-centricity.

The optical spatial resolution is checked with regard to the angle of incidence, lens aperture, the test pattern location in the camera field of view, the test patterns lighting and the distance between camera front and test pattern (working distance). Similar to the spatial calibration, no influence of the observation angle on the spatial resolution was present, besides the effect of varying working distance over the field of view in case of non-perpendicular observation. The working distance is varied between 177 mm and 185 mm for perpendicular observation, with a nominal working distance of 180 mm. The spatial resolution is determined by checking which elements of the test pattern the camera are still separable by eye. The best spatial resolution is achieved at full aperture with

a maximum of 19.7 + 0 − 2.1 μm at a 180 mm distance between camera lens front and test pattern. Only negligible resolution degradation occurs between a 180 and 183 mm distance (Figure 4). The manufacturer's statement of a spatial resolution of 10 μm at reduced aperture opening was not achieved, probably because the lighting was insufficient at minimal lens aperture and camera noise becomes dominating at low light conditions (see Figure 5a). Unfortunately, the application in an ion-beam analysis chamber prohibits more intense lighting due to a possible damage of the light sensitive radiation detectors.

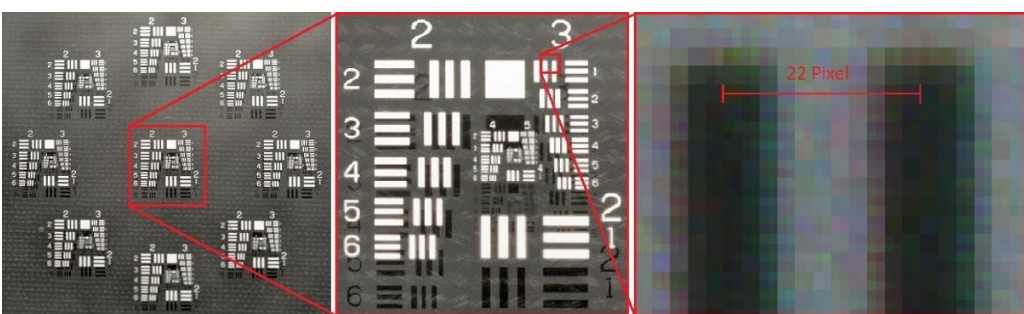

**Figure 4.** 1" × 3" Positive, USAF 1951 Wheel Target with Chromium lines on scintillating float glass. Each block contains a set of triple lines with decreasing size. The line distances and line width of each triplet are known, thus the smallest resolvable line triplet yields the camera or ion beam resolution. The right image shows a pair of lines with 22 pixels center to center distance. According to the specifications, there are eight pairs of lines per mm on this triplet. Therefore, a pair of lines has a distance of 0.125 mm and the pixels are calibrated to a spatial width of 5.68 μm.

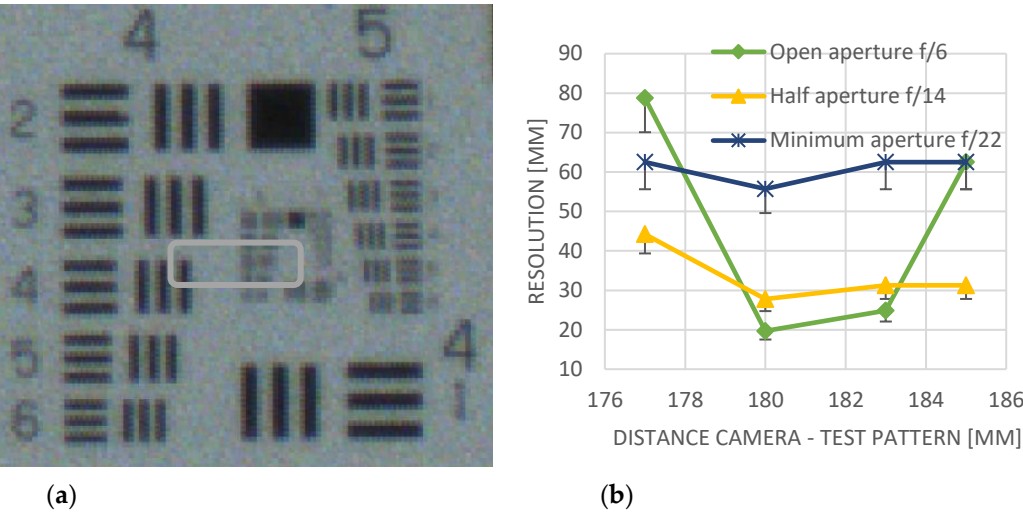

(**a**)  (**b**)

**Figure 5.** (**a**) Detail of the USAF 1951 test pattern with f/6 aperture. In the best case, the lines of the fifth element of the fifths group (box) can be separated, giving a spatial resolution of 19.7 μm. (**b**) Resolution with respect to aperture and working distance.

## 6. Magnetic Focusing System

A micro-beam has to provide a maximum ion current at a minimum beam diameter for high lateral resolution ion-beam analysis. For this purpose, beam focusing and beam apertures are required. The Quadrupole triplet consists of three OM-56 type magnets from the manufacturer Oxford Microbeams Ltd. (Oxfordshire, UK) with 10 mm bore and 100 mm length per magnet. The triplet is set at 123 mm distance between the last magnets iron core and the sample (see Figure 6). The magnets have a maximum field at the pole tip 0.4 T. They can be aligned with 1 μm precision. The beam current is specified to 0.6 nA for a spot size of 1 μm at a distance of 170 mm between last magnet and target by the manufacturer of the magnets. Practically, detection limits and accuracy for sub-μm depth resolution

requires ion currents >0.1 nA in order to achieve sufficient counts within measurement times in the order of hours which can be provided by the magnetic focusing system.

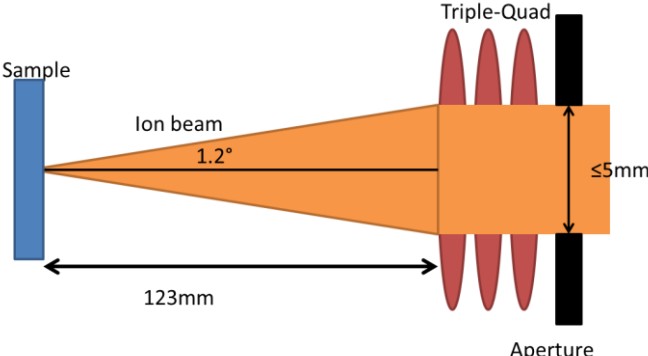

**Figure 6.** Sketch of the focusing setup. The ion beam originates from the right side. The object aperture defines its size. Three quadrupole magnets focus the beam onto the movable sample.

Since the micro-focus in this setup is necessarily connected to a convergent beam, an angular spread of the impacting ions of up to 1.2° is present for the maximum specified demagnification and a 5 mm aperture, see Figure 6. This geometrically induces ranges of probing depth and energy-loss of the projectiles in the target. Typically, smaller apertures and less demagnification is required for the investigated samples resulting in values of ≤0.3° for most cases. This value and the resulting loss of energy resolution is negligible compared to the geometrical straggling induced by the finite detector's width.

Two high-stability power supplies with up to 3 V and 100 A with <10 mA ripple and <5 ppm drift power the magnets. Connecting the power supplies in different polarity and grouping allows producing vertically elongated, horizontally elongated, or symmetric ion-beam spots. In general, it has to be considered that, due to the axial length of each magnet of 100 mm, the maximum demagnification on the sample position is achieved with quite different current at each magnet. To compensate for this, the central magnet is powered in opposite polarity to the two outer magnets and the final magnet is powered individually. The magnets can be spatially aligned on the scale of ±1–2000 μm vertically and horizontally and <5° rotation around the beam axis with respect to the initial adjustment of their baseplate by adjusting individual micrometer screws at each magnet base. Before operation, all magnets are centered on the beam axis by individually connecting each magnet in both polarizations and reducing their horizontal and vertical steering and the rhomboid focus effect to zero with the help of these micrometer screws. The beam spot on the sample is observed using the sample observation camera and a scintillating $LiAlO_2$ single crystal target irradiated by a 3 MeV proton beam of about 1 nA.

For the final step of setting up the focusing magnet currents, different groups of the USAF 1951 test image (Figure 5) are targeted with the sample manipulator discussed in the next section. The magnet currents were increased and the beam fluorescence size is compared to the known line width of the test image groups. The other beam parameters remain constant. Smaller line groups are targeted until no light can be observed next to the metallized part. This procedure is repeated for horizontal and vertical line groups, leading to an accuracy in the beam size determination of about 5% due to the finite inter group differences. The procedure allows determining the beam size, even for beam spots smaller than the optical resolution of the camera and for high current density beams. The minimum spot size is determined to $50 \times 35$ μm$^2$ with an object aperture of $1200 \times 1200$ μm$^2$. Correspondingly, a demagnification of 24 and 34 is achieved, respectively. Smaller spot sizes can be achieved by narrowing the beam apertures resulting in smaller beam currents depending on the emittance and current density of the connected accelerator. The beam-optical aberrations, the sample to magnet distance, and the accelerator beam brightness limit the minimum achievable beam spot size.

### 7. Sample Stage

A remote controlled in-vacuum manipulator (see Figure 2) allows for mapping the samples. This setup provides several advantages over ion-beam scanning, as a spatial calibration is not required, the ion beam can always stay in the ideal beam axis, a smaller magnet-sample distance (no scanning magnet) and a larger scanning area are achieved. The manipulator features 3 linear movement and 1 vertical rotational axis, all driven by piezo-electric engines. The rotational axis enables extended sample selection and an adjustment of the projectile impact angle, e.g., for channeling features with single crystals or gracing incidence analysis. The manipulator offers manipulation ranges of $\pm 25$ mm in linear axes with a positioning accuracy of 10 nm and a position resolution of 1 nm and an infinite rotational axis. The manipulator employs non-magnetic materials and engines with a dynamic range in movement of about $10^6$ and movement speeds of 1 nm/s to 10 mm/s.

A significant handling advantage is the overloading tolerance of the piezo engines, preventing damage to the engine when too much force is induced during mounting. The manipulator is calibrated and referenced by its internal positioning sensors, which were further verified by the tele-centric camera. The manipulator can handle weights of 300 g to 600 g, limited by the capabilities of the motor ($\pm 150$ g engine force), including the sample holder. This weight tolerance range was selected for the typical samples, but it can be tailored to other requirements by changing the counter-acting spring of the manipulator. On the manipulator, different types of sample holders can be installed on a self-centring adapter to fit either many small samples or fewer and larger samples. A so-called carrousel holder offers 40 positions for $10 \times 12 \times 5$ mm$^3$ PSI-2 [9] standard samples. The rotational axis enables selection of the 8 sample slots in this holder type. The samples are fixed in a profile avoiding contact of the front surface. A flat plate holder features two slit-like slots of 50 mm height and 5–10 mm width for clamping of samples between the plate and a backside spring-supported bar. This method conserves the sample to detector geometry with varying sample thickness. Alternatively, the sample thickness has to be known and corrected for by displacing the holder equally.

The rotational and tilt angles of the manipulator are aligned using screws for the tilt and a correction factor for the rotational axis. Their offset is determined by laser reflection from a mirror-finished sample mounted on the sample holder. For this purpose, a laser beam is injected into the back of the switching magnet. The laser beam travels through the 5 m long ion-beam line through a 2 mm aperture onto the sample chamber and is then reflected back from a mirror mounted on the sample holder. The manipulator tilt and rotation angles are corrected until the laser reflects back through the 2 mm entrance hole of the ruler at 100 mm distance. This leads to an angular accuracy of the alignment of $\pm 0.2°$. Therefore, the ion-beam impact angle has the same accuracy, but the detector has an additional uncertainty stemming from the relative alignment uncertainty of the ruler and the detector holder of $\pm 1$ mm. These tolerances in the ion-beam impact angle can lead to deviations of the Rutherford cross-sections in the order of 3%. The deviations of other cross-sections might be smaller and a general statement for the result uncertainty related to geometrical uncertainties is impossible. RBS generally features the lowest uncertainties among the four methods used here, due to the known and high cross-sections. Combining their results yields a total uncertainty larger than the largest individual uncertainty due to the rules of error propagation. Therefore, we state a minimal uncertainty of 3% for the results of the presented setup.

### 8. Ion Beam Analysis Detector Setup

The setup contains in total four detectors as depicted in Figure 7. For charged particle detection for nuclear reaction analysis (NRA) and backscattering spectrometry (RBS and EBS), two silicon detectors are installed at reaction angles of $150 \pm 2°$ for separating different particle species. The detector surface has a distance to the analysis/beam spot of $32.5 \pm 0.5$ mm. Light induced noise in the detectors is avoided by a full coverage of all windows during measurement. A high-resolution detector with 11 keV FWHM at 1.6 μs

rise-time (or 16 keV at 0.6 µs), 50 mm$^2$ size, and 100 µm nominal active thickness detects the full energy of protons up to about 3.3 MeV and α's up to about 13 MeV. Figure 8 shows the excellent resolution of this detector via the separation of Fe and Cr signals in a steel. The actual thickness of this detector is around 300 µm as full energy measurements of 6 MeV protons from NRA reactions suggest. The second detector has a 1500 µm nominal active thickness, 150 mm$^2$ size, and 22 keV FWHM resolution at 1.4 µs rise-time. This detector is equipped with a stopping foil of Polyimide of 25 µm thickness coated with 30 nm Al to block incident light and α-particles below about 5 MeV. Both detectors can be equipped with 2 mm thick steel apertures in the shape of a segment of the 150° circle to adapt count rates and limit geometrical straggling. The effective detector solid angle ratio between both detectors and different aperture options are experimentally calibrated using simultaneous acquisition of the $^7$Li(p, α)$^4$He reaction signal of a thin lithium containing film on a silicon substrate to an accuracy of 1% (>10,000 counts) for all cases with e.g., 3.52 ± 0.04 times reduced solid angle for a 1.5 mm aperture on the high-resolution detector and an open thick detector. The connected RC pre-amplifiers limit the usable count rates up to a few 10,000/s with significant degradation of energy resolution at higher values.

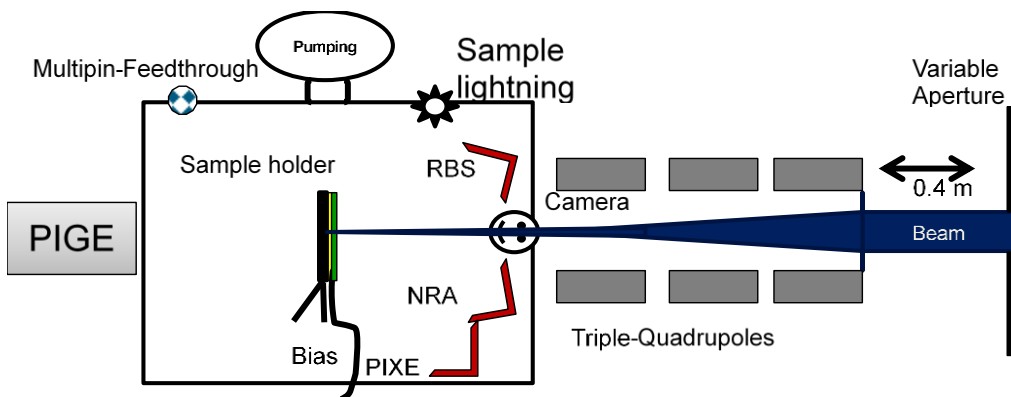

**Figure 7.** Sketch of the detector and chamber arrangement.

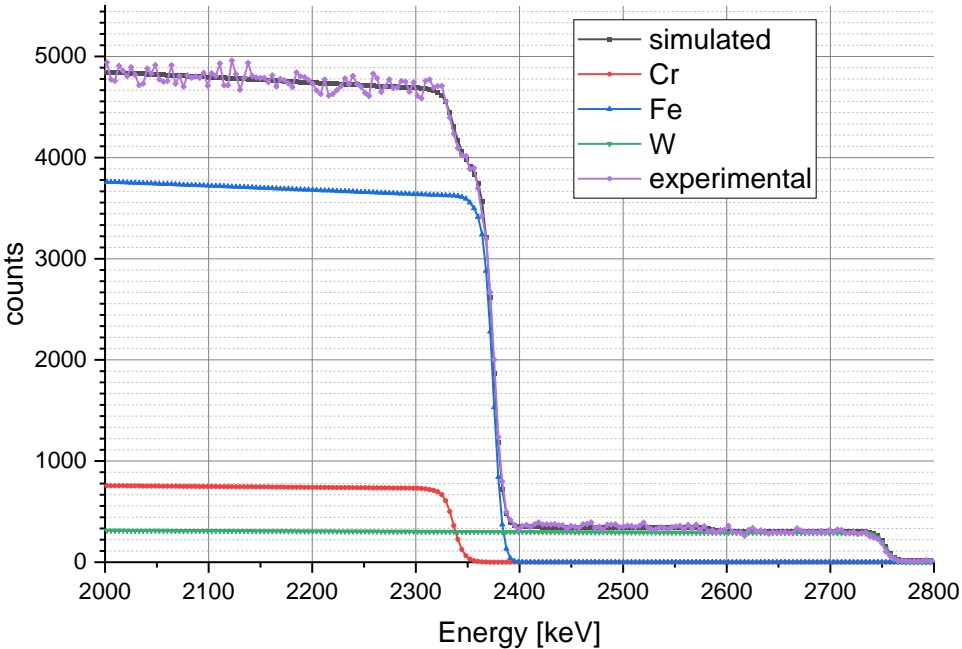

**Figure 8.** RBS detector signal using 2960 keV $^3$He ions on HiperFer 17Cr steel with <2% dead-time together with SimNRA evaluation. The setup allows for a separation of the Fe and Cr signals with a clear edge at ~2350 keV, yielding a matching Cr concentration in the steel.

For particle induced x-ray analysis (PIXE), a KETEK AXAS-D silicon-drift detector of 450 µm active thickness and an 8 µm thick Beryllium foil counts the emitted X-rays. The detector allows for count rates up to 200,000/s for <10% deadtime and <135 eV resolution. The detector normal points at the sample surface with an angle of 62.5° from a port in the upper plane. Its distance to the sample can be adjusted from 30 to 200 mm to optimize count rates. The detector is calibrated by K and L lines from high purity W, Mo and Y targets. Its H factor (energy dependent sensitivity) is determined by the same samples in combination with normalization by the respective RBS signals.

A HPGe detector with 76.2 mm crystal diameter is installed at the backside flange of the chamber as seen by the ion beam for particle induced gamma-ray analysis (PIGE). In this geometry, the photons will be detected only after passing the sample and the sample holder, but for the relevant photon energies and the holder construction, only a minor absorption can be expected. For the efficiency calibration, known lithium, aluminum, and boron samples are exposed together with RBS and NRA measurements for cross-calibration. The system achieves an overall detection efficiency in the order of 1% (478 keV photons).

A quantitative analysis of the sample composition by ion beam analysis requires determining the Particle*Sr value (=incident ion dose * detector solid angle). This value is the product of detector solid angle and incident ion charge/dose. The ion current is measured using a Keithley 6487 pico-ampere meter and integrated by software to a precision <0.1%. The actual relevant (systematic) uncertainty in the collected charge arises from the emission of secondary electrons (SE) by the ion beam impact. For suppressing these SE and measuring only the ion current, an electric field driving the SE back onto the sample is required. In the current setup, the sample holder is isolated against the surrounding structures and the (metallic) sample holder itself is biased to $-10$ to $+500$ V against the ground using the voltage sources of the pico-ampere meter. When adding a conductive tape to the sample front and operating with <nA currents, even diamond with their high electrical resistance could be measured without arc discharges of the deposited charges in this setup. The flexible biasing scheme allows a maximum suppression of SE, as it acts against all directions ($4\pi$), but also an exploitation of the SE for analytical/imaging (SEI) purposes. The suppression of SE for a correct measurement of the incident ion current is tested using a 1.4 MeV $^4$He$^+$ beam on the aluminum sample holder and 2.96 MeV $^3$He$^+$ on a plasma deposited hydrogenated carbon layer on graphite with the results shown in Table 2. The sample current is measured for different sample bias voltages. The measured current stabilizes above +60 V, indicating a complete suppression of the secondary electrons for these beam parameters, but the ion beam current stability was insufficient for a remaining SE contribution of <2%.

**Table 2.** Measured sample current for two different ion beam settings on different materials. In both cases, the current increases with negative and decreases with positive biasing of the sample holder. At around +80 V bias, the secondary electrons seem to be completely suppressed in both cases.

| Bias [V] | −10 | −4 | −2 | −1 | 0 | +1 | +3 | +10 | +30 | +50 | +60 | +80 | +100 |
|---|---|---|---|---|---|---|---|---|---|---|---|---|---|
| 1.4 MeV $^4$He$^+$ [nA] | 13.1 | 12.2 | 11.4 | 10.4 | 9.1 | 7.7 | 5.8 | 3.1 | 2.4 | 2.28 | 2.22 | 2.22 | 2.22 |
| 2.96 MeV $^3$He$^+$ [nA] | 9.1 | 8.4 | | | 7.2 | | 3.2 | 2.6 | 1.65 | 1.5 | | 1.4 | 1.4 |

## 9. Conclusions

A new setup for ion-beam analysis on the µm scale is developed, tested, and evaluated. The setup combines state-of-the-art industrial products to a reliable device with easy handling, high precision, and high throughput in the order of 40 samples or about 3000 points on one sample per day, as already demonstrated in [4]. The setup consists of four independent parts, the vacuum system, the magnetic focusing, the sample observation and positioning, and the detectors for ion-beam analysis. Special attention is paid on maintaining a vacuum below $10^{-7}$ mbar without inducing stray magnetic fields and vibrations into the sample chamber. In the sample chamber, a combination of a fixed beam axis with an

nm precision piezo-electric manipulator and a high-resolution tele-centric camera proved to be reliable and practical for sample positioning. Four detectors for charged particles, $\gamma$-rays, and X-rays with special apertures and stopping foils enable a Total-IBA analysis with high depth resolution, sub-% accuracy, and ppm detection limits using 0.01 to 50 nA ion beam currents. The good energy resolution in combination with the low geometrical straggling enable a surface near separation of Fe and Cr in steels using RBS. Generally, the RBS and NRA detector count-rates limit the performance of the presented setup due to excessive dead-time or loss of energy resolution in the connected amplifier electronics.

The installation and adjustment of the device posed a particular challenge, since sub-mm and sub-degree precision is required in the alignment of the different parts. The chosen compact arrangement improves the analytical performance, but also increases the requirements on accuracy. Several steps using levels, leveling cameras, lasers, and adjusting screws with two stages of accuracy on several spots yielded success in alignment. The setup precision in angles and distances is sufficient to deliver 3% accuracy in the ion beam measurements. The final step required alignment of the magnets on the $\mu$m scale with direct ion beam observation and allowed to obtain the specified demagnification of about 24. The use of a test pattern on (scintillating) float glass allowed the determination of beam position and size.

The connected tandem accelerator from 1984 limits the achievable beam spot sizes to a few 10 $\mu$m, but a modern accelerator would allow for beam spots in the order of 1 $\mu$m due to increased beam brightness. For the future, several projects on depth analysis with side cuts, grain resolved studies, ppm sensitivity elemental analysis, and in-situ experiments of battery cells and a commercial product based on this design are planned. As an experimental setup, the technical details and implemented devices will change and improve over time, but the principle choice in the design space will remain.

## 10. Patents

The patent WO002019219103A1 relates to this publication: Sören Möller [1,*], Daniel Höschen [1], Sina Kurth [1], Gerwin Esser [1], Albert Hiller [1], Christian Scholtysik [2], Christian Dellen [1], and Christian Linsmeier [1].

**Author Contributions:** Conceptualization, S.M., D.H., and C.L.; methodology, S.M.; software, S.M.; investigation, S.M., S.K., and C.D.; resources, C.S., G.E., A.H., and C.D.; data curation, S.M.; writing—original draft preparation, S.M.; writing—review and editing, S.M.; visualization, S.M.; supervision, C.L.; funding acquisition, C.L. All authors have read and agreed to the published version of the manuscript.

**Funding:** This research received no external funding.

**Data Availability Statement:** Data sharing is not applicable.

**Conflicts of Interest:** The authors declare no conflict of interest. The funders had no role in the design of the study; in the collection, analyses, or interpretation of data; in the writing of the manuscript, or in the decision to publish the results.

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
