# Peer review of "A New High-Throughput Focused MeV Ion-Beam Analysis Setup"

_instruments, doi:10.3390/instruments5010010_

Round 1

Reviewer 1 Report

In this manuscript, a new focused MeV ion-beam analysis setup installed in Forschungszentrum Jülich was described. No new physics included in this work, only a very detailed, careful description of this setup was given.

The manuscript is well structured, free from errors and fit to the aim and scope of the journal Instruments. The work will be of interest to the related materials science community, first of all the users of this setup, they can cite this paper where the detailed experimental conditions are given. Thus, I suggest acceptance of the manuscript after a minor revision. The following points should be addressed:

  • page 9, line 322: “The manipulator features 3 linear movement and 1 vertical rotational axis,” compare to line 336: “The rotational and tilt angles of the manipulator”. How many axes are there in the manipulator?
  • Is it possible to use channelling effect during IBA experiment for single crystalline sample? If not, how partial channelling can be avoid?
  • Please, give the ranges of tilt and rotation angles. Is rotation used only to change the samples in carrousel holder?
  • To the precise measurements, it is significant that the tilt axis of the manipulator has to be exactly on the sample surface in order to the beam remaining on the same spot with changing the tilt angle. How it is ensured for various sample thicknesses?
  • Is it possible to perform IBA analyses if the samples are lightened during IBA experiment, i.e., under the beam?
  • Is it possible to perform IBA on insulator samples using this setup? How can the charging up processes be avoided? How can the current be determined?
  • Please, describe the limitation of the setup, too.

Author Response

Dear Reviewer 1,

thank you for your review and comments. All suggestions were implemented, see replies below.

  • page 9, line 322: “The manipulator features 3 linear movement and 1 vertical rotational axis,” compare to line 336: “The rotational and tilt angles of the manipulator”. How many axes are there in the manipulator?
    ->The formulation was made clearer, line 336 talks about alignment before operation
  • Is it possible to use channelling effect during IBA experiment for single crystalline sample? If not, how partial channelling can be avoid?
    -> in principle yes, but the device was not optimised for this task. a statement was added. The rotational axis enables avoiding channeling by . The limitation to one rotational axis limits the use for channeling analysis
  • Please, give the ranges of tilt and rotation angles. Is rotation used only to change the samples in carrousel holder?
    -> a statement was added
  • To the precise measurements, it is significant that the tilt axis of the manipulator has to be exactly on the sample surface in order to the beam remaining on the same spot with changing the tilt angle. How it is ensured for various sample thicknesses?
    -> the hardware is not built like this, but with the given linear axes it can be compensated by software. Sample thicknesses have to be known (carousell holder) or samples mounted with their front side. A statement was added.
  • Is it possible to perform IBA analyses if the samples are lightened during IBA experiment, i.e., under the beam?
    -> You mean for scintillating samples or if the light is switched on? In principle yes, the only detector sensitive to light is the RBS detector. Here additional noise is induced by the light. It depends on your analysis goals and how much light you need if that is acceptable or not.
  • Is it possible to perform IBA on insulator samples using this setup? How can the charging up processes be avoided? How can the current be determined?
    -> Yes it is possible. Usually a surface contact with carbon tape is enough for sub-nA currents. Diamond was successfully measured with these conditions. A sentence was added
  • Please, describe the limitation of the setup, too.
    ->a sentence was added regarding detector performance and spot sizes. The limitation of the connected accelerator was already present.

Reviewer 2 Report

General :

The manuscript by S. Moller et al., “A new high-throughput focused MeV ion-beam analysis setup” describes in detail a new setup for performing ion-beam analysis via various techniques. The new setup seems very interesting, and despite its challenging installation, stems from the paper that focus on details has been given in order to improve its performance. At the same time, and despite the fact that the authors have described in great detail the technical details of the various instruments, the detectors and the precision (including systematics), the paper lacks coherence and is very difficult to follow through. Most importantly, it lacks a clear description of a) the *novelty* of the new line compared to other existing installations b) The capabilities of the setup in terms of species, current, beam-momenta etc… and c) examples of successful experiments or even commissioning results in order for the reader to understand the choices of the authors on the technical level. In its present form, the article looks rather a comprehensive lab report rather than a journal article on physics instrumentation. This is also stated by the authors in line 21, of the abstract.

I recommend a major revision of the article before proceeding to consider publication. The authors need to reconsider the article’s structure, first describing the injector chain, then the beam-line and then the sample holder + detectors in a clear and coherent way.

Detailed comments line-by-line can be found below. Among those there are some terminology issues, as well as English/grammar language points that need to be addressed.

Title: The title is too long and confusing. How about: A novel, high-precision ion beam analysis setup?

Abstract:

Line 8: Add a comma after exists “…A pool of IBA methods exists, from which…”

Line 9:  Provides à Offers

Line 13: High throughput and accuracy à Of what? Please precise. Also in the next line, “Tolerances” à Please specify. Positioning? Something else?

Line 13-14: “enables pressures of 5x10^-8 mbar” à “Allows vacuum levels down to 5*10^-8 mbar”

Line 18: “up to 3000 points per day”. Please specify what do you mean by “points” ?

Line 19: “Special apertures”-> “Custom-made apertures”

Line 20-21: The last sentence compromises the novelty in my opinion. In the paper the novelty of the setup should be demonstrated, and what really makes it unique. Technical details could be published in a lab-report or an internal note.

Introduction:

General:  Throughout the paper it must be clear if the energy is total energy (or kinetic)  and if it is per A or per Z.

Line 28: Remove “reference free”.

Line 31: “For this reasons” -> For these reasons

Line 35: “Mostly” -> “Usually” and remove “chambers”.  The measurements are conducted in vacuum, due to the important energy loss of ion beams on air.

Line 37: This -> These

Line 41: What unambiguous result ? briefly mention. E.g like in the case ….You cannot expect the reader to read the whole reference [2].

Line 49 : Reference for the important application of battery research missing.

Line 51 : small beam diameter -> I suppose you would like to write rather “beam emittance”. Line 52: “good counting statistics” à decent statistics

Line 53: Rephrase : “These contraditctions are brought together by ion beam focusing” -> “This can be achieved with proper beam optics that take into account the initial emittance from the accelerator and the betatron oscillations in the ion transfer-line”.

Line 55: You refer  to reference [7] ? Remove “instrument”. Line 57, why potentially ? The authors of [7] claim that they achieved this.

Line 60: Not clear where the authors refer to. To their work ? To reference [7]  ? Please clarify.

Main body:

General: From here it is really missing a brief general introduction on the layout of the accelerator complex. The authors try to show it in Figure 1, but it should be stated before and foremost. Where the ions originate ? Are possible species possible ? What is the energy of the accelerator ? What type of extraction ?  Beam current ?  A discussion on radiation is also very important.

Lines 65 – 73: Serious rephrasing and coherence is missing from this paragraph. The authors start discussing “limitations”  of some (?) system while the mu-NRA system can solve them….? Rephrase necessary of the old / other setups and the author’s proposal for the new setup should be given.

Figure 1: The gray boxes inside the mu-NRA are not clear. The same stands for various elements in the accelerator. Please describe in the Figure’s caption all the elements. What are the lines inside the “accelerator” ?

Figure 2: The reader should understand that now they look at a CATIA drawing of the final part of the line ? It is not at all clear how Figures 2 (a) and (b) combine. Can you add a drawing of the complete setup and in relation with the Figure 1 ?

 Line 90: The authors mean that the turbo and the fore pumps are identical ? Where is their position ?

Line 91: In the introduction the authors mention 5x10^-8 bar, while here 5*10^-9. For the Faraday caps and the monitor mention the X0 (radiation length). Do they have important effect on the beam shape / divergence ?

Line 94: The role of the apertures is not at all clear. Is it cleaning ? Is it selection of phase-space ? Why are they placed at this point ? Or maybe this is discussed later, in the optics ? Better use also the more well-known term “collimator”.

Line 96: Low radiation levels. Where ? On the samples? On the environment ? What is “low” ?

Line 102: “Type K” -> type “K”. Which brand ?

Line 104: Kapton -> kapton

Figure 3: Put labels to show which are the batteries. Why the different coloring ? The wiring cannot be clearly seen. Indicate it with an arrow ?

Line 111: “During assembly” -> “During THE assembly”

Lines 112-113: Please rephrase as “in order to minimize the disturbance in the beam properties”. Again here, not knowing the exact energy of the beam is difficult for the reader to understand the effect. Are there any simulations demonstrating how strong this effect is ? It would be great if a map demonstrating the numbers that the authors claim could be provided.

Line 130: Article missing. “After the beam line and chamber” assembly

Line 134: Why the beam is drifting ? The temperature on the bending magnets affects the field ? The authors should elaborate on the reasons of this drift. Moreover, we are talking about um ? mm ? Give an order of magnitude.

Line 143: “For long-term ion beam measurements” -> “For experiments requiring longer data-acquisition times”.

I have a serious difficulty to understand what the message by the authors in these lines is. What do they mean that the positional difference is the relevant quantity ? For what ? This paragraph (lines 143- 152) needs to be re-written and understood better.

Line 156: Give an order of magnitude for the pump-down times.

Line 206: Colour -> color

Line 256: The resolution of the camera is only a function of its focal properties -> Therefore I am a bit surprised by the author’s statement that the insufficient lighting. Can you please clarify further ?

Line 267: For first time in the manuscript, the authors introduce the term “microbeam”. This is a term used by their reference [7] and should be clearly mentioned in the introduction that will be used from now on as such.

Line 269: “A triple-quadrupole magnet consisting of three oxford Microbeams OM-56”.  I don’t understand. Do the authors mean “a quadrupole triplet”, constructed by “Oxford Microbeams” company, model OM-56 ?  Are those electrostatic quadrupoles ? The authors mention “10 mm bore” --> Better use the more common word “aperture”.  Quadrupole length and maximum gradient (T/m) ? They mention “0.4 T on axis” (is this the integrated gradient T/m *m ) ?! Also “induce” is not a correct word, I would say “they have a peak integrated strength of …”

Line 270: Again, the setup is not clear until one looks in Figure 6. Rephrase: “The distance from the end of the last quadrupole to the sample was set to 123 mm”.

Line 272: I do not understand why the *beam* current (0.6 nA) as stated by the authors is relevant with the focusing distance of the quadrupoles. I seem to be missing something. The image-size is a function of the triplet configuration. What are the triplet polarities ? FDF?  DFD ? FFD or different ?

Lines 274-277: Remove. They do not add any new information and compromise clarity.

Lines 278 -282: Again, not at all clear. What do you mean by “demagnification” ? Do the authors refer to the optical term (cosine-like ray)  ? Or you mean, in geometrical terms, that the maximum beam angle should be specific in order to allow focus at a certain point ? But this is a function of the triplet configuration, right ? Again : What are the triplet optics for this configuration ? Please show the optical functions or at least state clearly the focal lengths in various cases. Do you refer to horizontal plane / vertical plane or both ?

Line 285: I presume that the authors mean that with different settings on the quads they can produce assymetric or symmetric beams. Can you provide a few simulated plots of the spot-size, or at least give an order of magnitude in sigmas ?

Lines 286-296: I am completely lost here. Axial length è Longitudinal length of the quadrupoles? I suppose you refer to the yoke ? The term “excitation” is not very common for a magnet – use rather “Magnetic field strength”. Also, the authors claim that the magnets can be “adjusted”. In which direction? Longitudinally? Radially? I suppose longitudinally, since this is the parameter that affects the focal length of the lens system. The authors claim that the spot-size is “checked” by a proton beam of 3MeV/c and a current of 1nA. I suppose that he currents are scaled for higher momenta? This is not clear and has to be mentioned. In line 298 a “manipulator” is mentioned, which has not been introduced before causing confusion.

Line 306: Object aperture? You mean object dimensions? I suppose you refer to the aperture just before but this is a bit strange. Please clarify. Again here: Demagnification is confusing : I would say a negative magnification of 24 mm / mm in the horizontal plane and 34 mm / mm in the vertical plane.

Line 307: I don’t get how narrowing the aperture leads to larger spot-sizes. Again, it is quite confusing what do the authors mean here and how the current is connected with the spot-size and the accelerator emittance?!! Maybe I have misunderstood something here.

Line 315: Add a reference here for Figure 2. The “manipulator” is a remote high-precision stage ?

Line 326-327: “The sample weight acceptance” -> “The maximum weight of the samples that can be placed on the manipulator is”

Line 331-335: Too many details here. The authors could just cite reference [8] instead of discussing the different types of sample holders that bring no added value to the paper.

Line 338: For first time we learn that the length of the line is 5 meters. As mentioned above, a clear schematic of the beam-line should be put in the *introduction*.

Lines 357-358: I do not understand. The thickness of the detector is not known, and the authors have measured it with the beam?!!

Line 370: What is the role of the foil ? Please explain. Also: The detector does not “look” at the samples, is placed with respect to the samples at an angle of 45 degrees.

Line 385: What is Particle *Sr ? Please define.

Line 387: What does it mean “integrated by software” ? You mean that the counting is done on the software level with an intrinsic error of 0.1% ?

Table 2: Ion beam settings (not setting). Also for first time here we learn that the species are 3-He+ and 4-He+, non-fully stripped. The X0s on the beam-line do not strip these ions ? This should be clearly discussed/ explained in the introduction.

Line 406: The authors claim that the new setup for ion-beam analysis is “developed, tested and evaluated”. However from the whole paper, it rather seems like a pure description of the instruments and their characteristics. The authors do not evaluate, nor compare with similar setups, nor show any experimental plots to confirm the novelty of this new line. Also: No mention of the users, as well as examples of successful experiments. So the question comes, has the setup been tested ? Is it on the design phase ? It is operational ? These points should be addressed in the discussion, also clearly stating the *novelty* of this new setup with other existing setups in the world.

Author Response

Dear Reviewer 2,

thank you for your comments. Indeed the paper rather describes a new device than a full scientific novelty. In fact another reviewer had the same comment, but derived a positive recommendation for publication from this. Two example applications were cited and a few more exist. The sample throughput was the main goal and a reason for most technical decision. This requires high count-rates, easy handling, maximal scanning volume and automatisation. If this didnt come out clear enough, we will try to reformulate parts of the text.

Most comments were implemented in connection with the other 2 reviewer comments. Please find the detailed replies below.

Title: The title is too long and confusing. How about: A novel, high-precision ion beam analysis setup?

the high throughput aspect is of particular relevance in my field so i would like to keep this.

Abstract:

Line 8: Add a comma after exists “…A pool of IBA methods exists, from which…”

added

Line 9:  Provides à Offers

the device exists now so i would say it offers and provides

Line 13: High throughput and accuracy à Of what? Please precise. Also in the next line, “Tolerances” à Please specify. Positioning? Something else?

words added

Line 13-14: “enables pressures of 5x10^-8 mbar” à “Allows vacuum levels down to 5*10^-8 mbar”

wording changed

Line 18: “up to 3000 points per day”. Please specify what do you mean by “points” ?

the method analyses in discrete spatial points for which a certain time is required for a given accuracy

Line 19: “Special apertures”-> “Custom-made apertures”

changed

Line 20-21: The last sentence compromises the novelty in my opinion. In the paper the novelty of the setup should be demonstrated, and what really makes it unique. Technical details could be published in a lab-report or an internal note.

ok, this is a question to the editors. In my understanding "instruments" covers this type of article.

Introduction:

General:  Throughout the paper it must be clear if the energy is total energy (or kinetic)  and if it is per A or per Z.

its always kinetic energy without normalisation

Line 28: Remove “reference free”.

within the ion beam analysis community this is considered a major feature and derives from the knowledge of the Rutherford cross-section.

Line 31: “For this reasons” -> For these reasons

corrected

Line 35: “Mostly” -> “Usually” and remove “chambers”.  The measurements are conducted in vacuum, due to the important energy loss of ion beams on air.

changed

Line 37: This -> These

corrected

Line 41: What unambiguous result ? briefly mention. E.g like in the case ….You cannot expect the reader to read the whole reference [2].

added a sentence with an example

Line 49 : Reference for the important application of battery research missing.

i hoped it to be published by now, but its not. removed statement.

Line 51 : small beam diameter -> I suppose you would like to write rather “beam emittance”.

no, in this case the beam diameter counts. a shorter focal length can produce a smaller beam diameter for a given emittance.

Line 52: “good counting statistics” à decent statistics

changed

Line 53: Rephrase : “These contraditctions are brought together by ion beam focusing” -> “This can be achieved with proper beam optics that take into account the initial emittance from the accelerator and the betatron oscillations in the ion transfer-line”.

In fact betatron and emittance were not taken into account here, so i dont want to mention this. The accelerator is simple and only a few meters long with 2 dipoles and a few MeV. We dont have the knowledge of the beam properties, except for a rough idea of size and the beam energy. the layout is done on an estimation basis.

Line 55: You refer  to reference [7] ? Remove “instrument”. Line 57, why potentially ? The authors of [7] claim that they achieved this.

the new device achieves a similar demagnification as the one presented in [7], but in [7] a higher brightness beam was available. With a modern ion source we could achieve the same µm spot size, but since we dont have this equipment attached to our device we cannot achieve this spot size.

Line 60: Not clear where the authors refer to. To their work ? To reference [7]  ? Please clarify.

sentence clarified

Main body:

General: From here it is really missing a brief general introduction on the layout of the accelerator complex. The authors try to show it in Figure 1, but it should be stated before and foremost. Where the ions originate ? Are possible species possible ? What is the energy of the accelerator ? What type of extraction ?  Beam current ?  A discussion on radiation is also very important.

Details were added. Ion beam analysis applies ions from H to Au and we use beam energies from 0.5 to 5 MeV.

Radiation is discussed later with regards to aperture materials. Despite this it mostly depends on the analysed sample, which varies. A discussion of an application example can be found in [4] together with a discussion of the radiation scenario.

Lines 65 – 73: Serious rephrasing and coherence is missing from this paragraph. The authors start discussing “limitations”  of some (?) system while the mu-NRA system can solve them….? Rephrase necessary of the old / other setups and the author’s proposal for the new setup should be given.

paragraph rephrased

Figure 1: The gray boxes inside the mu-NRA are not clear. The same stands for various elements in the accelerator. Please describe in the Figure’s caption all the elements. What are the lines inside the “accelerator” ?

caption extended. The lines in the accelerator are the acceleration structures/rings

Figure 2: The reader should understand that now they look at a CATIA drawing of the final part of the line ? It is not at all clear how Figures 2 (a) and (b) combine. Can you add a drawing of the complete setup and in relation with the Figure 1 ?

caption extended.

 Line 90: The authors mean that the turbo and the fore pumps are identical ? Where is their position ?

no, the combination is identical between chamber and beamline. The positions are shown in fig.1

Line 91: In the introduction the authors mention 5x10^-8 bar, while here 5*10^-9. For the Faraday caps and the monitor mention the X0 (radiation length). Do they have important effect on the beam shape / divergence ?

5e-9 is the beamline, 5e-8 is the sample chamber. Fcup and monitor description extended. What is X0? Is it the range of ions in matter? Here with the low beam energies this is <<0.1mm

Line 94: The role of the apertures is not at all clear. Is it cleaning ? Is it selection of phase-space ? Why are they placed at this point ? Or maybe this is discussed later, in the optics ? Better use also the more well-known term “collimator”.

It is phase space selection for limiting dimension and divergence. In the microbeam context collimator usually refers to the last aperture

Line 96: Low radiation levels. Where ? On the samples? On the environment ? What is “low” ?

description extended

Line 102: “Type K” -> type “K”. Which brand ?

changed. Its self-made from respective wires.

Line 104: Kapton -> kapton

no, Kapton is a brand name. The material is polyimide

Figure 3: Put labels to show which are the batteries. Why the different coloring ? The wiring cannot be clearly seen. Indicate it with an arrow ?

indications added and caption extended. The batteries are covered with a copper contact. Upon charging lithium plates here yielding this white-blueish colour.

Line 111: “During assembly” -> “During THE assembly”

changed

Lines 112-113: Please rephrase as “in order to minimize the disturbance in the beam properties”. Again here, not knowing the exact energy of the beam is difficult for the reader to understand the effect. Are there any simulations demonstrating how strong this effect is ? It would be great if a map demonstrating the numbers that the authors claim could be provided.

We dont have any simulations. In fact it works like this. We added a vacuum gauge to the system and saw different values required at the steerer for the same spot position on the sample. The we added a distance tube to the gauge to increase its distance to the beam and we could again use the original (near zero) steerer values. From this we learned and mounted all magnetic parts at a certain distance. Sounds maybe like a car workshop if you are used to high energy physics labs, but i have done layout, design, order, assembly and operation of this device mostly by myself, with some help with CAD and the heavy parts. There are simply no resources for this type of analysis

Line 130: Article missing. “After the beam line and chamber” assembly

changed

Line 134: Why the beam is drifting ? The temperature on the bending magnets affects the field ? The authors should elaborate on the reasons of this drift. Moreover, we are talking about um ? mm ? Give an order of magnitude.

description extended. The order of magnitude follows in the next paragraph

Line 143: “For long-term ion beam measurements” -> “For experiments requiring longer data-acquisition times”.

changed

I have a serious difficulty to understand what the message by the authors in these lines is. What do they mean that the positional difference is the relevant quantity ? For what ? This paragraph (lines 143- 152) needs to be re-written and understood better.

paragraph improved

Line 156: Give an order of magnitude for the pump-down times.

added

Line 206: Colour -> color

i am trying to follow BE

Line 256: The resolution of the camera is only a function of its focal properties -> Therefore I am a bit surprised by the author’s statement that the insufficient lighting. Can you please clarify further ?

that depends on your limitation. Also electronic noise plays a role as you can see in fig. 5a. The S/N ratio depends on the lightning. Also the lens aperture can be reduced with more light, further improving resolution

Line 267: For first time in the manuscript, the authors introduce the term “microbeam”. This is a term used by their reference [7] and should be clearly mentioned in the introduction that will be used from now on as such.

microbeam added to the introduction

Line 269: “A triple-quadrupole magnet consisting of three oxford Microbeams OM-56”.  I don’t understand. Do the authors mean “a quadrupole triplet”, constructed by “Oxford Microbeams” company, model OM-56 ?  Are those electrostatic quadrupoles ? The authors mention “10 mm bore” --> Better use the more common word “aperture”.  Quadrupole length and maximum gradient (T/m) ? They mention “0.4 T on axis” (is this the integrated gradient T/m *m ) ?! Also “induce” is not a correct word, I would say “they have a peak integrated strength of …”

Sentence rearranged. We dont have data on the gradient available, sorry.

Line 270: Again, the setup is not clear until one looks in Figure 6. Rephrase: “The distance from the end of the last quadrupole to the sample was set to 123 mm”.

rephrased

Line 272: I do not understand why the *beam* current (0.6 nA) as stated by the authors is relevant with the focusing distance of the quadrupoles. I seem to be missing something. The image-size is a function of the triplet configuration. What are the triplet polarities ? FDF?  DFD ? FFD or different ?

a shorter distance allows for a smaller focal length and this reduces the beam dimensions (while increasing divergence) or allows to use a larger aperture opening for the same spot size. See my book [1] section 2.3.2

The details on the configuration are discussed later in the text.

Lines 274-277: Remove. They do not add any new information and compromise clarity.

shortened and clarified these sentences

Lines 278 -282: Again, not at all clear. What do you mean by “demagnification” ? Do the authors refer to the optical term (cosine-like ray)  ? Or you mean, in geometrical terms, that the maximum beam angle should be specific in order to allow focus at a certain point ? But this is a function of the triplet configuration, right ? Again : What are the triplet optics for this configuration ? Please show the optical functions or at least state clearly the focal lengths in various cases. Do you refer to horizontal plane / vertical plane or both ?

demagnification is the factor by which the beam spot on the sample is smaller than the aperture opening, so the geometrical size. Indeed it depends on the triplet configuration, but there is always a configuration for minimum size, but this depends on the applied ion beam species and energy. Both of them vary.

Line 285: I presume that the authors mean that with different settings on the quads they can produce assymetric or symmetric beams. Can you provide a few simulated plots of the spot-size, or at least give an order of magnitude in sigmas ?

no sorry, this kind of simulations exceeds our available resources. the only thing we have is the dimension measured by a scintillator and the camera. This size varies with beam species and energy

Lines 286-296: I am completely lost here. Axial length è Longitudinal length of the quadrupoles? I suppose you refer to the yoke ? The term “excitation” is not very common for a magnet – use rather “Magnetic field strength”. Also, the authors claim that the magnets can be “adjusted”. In which direction? Longitudinally? Radially? I suppose longitudinally, since this is the parameter that affects the focal length of the lens system. The authors claim that the spot-size is “checked” by a proton beam of 3MeV/c and a current of 1nA. I suppose that he currents are scaled for higher momenta? This is not clear and has to be mentioned. In line 298 a “manipulator” is mentioned, which has not been introduced before causing confusion.

description clarified. There is no radial direction here, its a linear accelerator.

The beam energy is 3 MeV, we dont use momentum is this context. Hence also no scaling, 1nA is simply the current measured on the sample.

Reference to following manipulator discussion added

Line 306: Object aperture? You mean object dimensions? I suppose you refer to the aperture just before but this is a bit strange. Please clarify. Again here: Demagnification is confusing : I would say a negative magnification of 24 mm / mm in the horizontal plane and 34 mm / mm in the vertical plane.

"Object" and "collimator" aperture are usual term in the microbeam ion beam analysis community for first and second aperture. The demagnification is a size factor of the beam at the aperture vs. at the sample.

Line 307: I don’t get how narrowing the aperture leads to larger spot-sizes. Again, it is quite confusing what do the authors mean here and how the current is connected with the spot-size and the accelerator emittance?!! Maybe I have misunderstood something here.

"Larger" exchanged for "smaller". The current density and the aperture size/area define the beam current on the sample.

Line 315: Add a reference here for Figure 2. The “manipulator” is a remote high-precision stage ?

added

Line 326-327: “The sample weight acceptance” -> “The maximum weight of the samples that can be placed on the manipulator is”

This is not exactly what is meant. The manipulator can bear more weight, it just wont move in the vertical axis then.

Line 331-335: Too many details here. The authors could just cite reference [8] instead of discussing the different types of sample holders that bring no added value to the paper.

The holder have an important aspect for the sample to detector distance, as this might change with sample thickness or not, depending on holder type. Other reviewers requested this information.

Line 338: For first time we learn that the length of the line is 5 meters. As mentioned above, a clear schematic of the beam-line should be put in the *introduction*.

Your statement is wrong, the length of 5m is specified already in line 90.

for style reasons i try to restrict the introduction to the general frame and the background and only start with the actual work in the following section. Its a question to the editors.

Lines 357-358: I do not understand. The thickness of the detector is not known, and the authors have measured it with the beam?!!

yes, via the energy loss of 6MeV protons. Detectors sold with a minimum active thickness specification are somewhat cheaper.

Line 370: What is the role of the foil ? Please explain. Also: The detector does not “look” at the samples, is placed with respect to the samples at an angle of 45 degrees.

changed

Line 385: What is Particle *Sr ? Please define.

explained in the following sentence

Line 387: What does it mean “integrated by software” ? You mean that the counting is done on the software level with an intrinsic error of 0.1% ?

yes, the uncertainty relates to the current measurement accuracy. there are also hardware implementations of charge integration which were not implemented here.

Table 2: Ion beam settings (not setting). Also for first time here we learn that the species are 3-He+ and 4-He+, non-fully stripped. The X0s on the beam-line do not strip these ions ? This should be clearly discussed/ explained in the introduction.

these species are only examples. ion beam analysis works with everything from H to Au

Line 406: The authors claim that the new setup for ion-beam analysis is “developed, tested and evaluated”. However from the whole paper, it rather seems like a pure description of the instruments and their characteristics. The authors do not evaluate, nor compare with similar setups, nor show any experimental plots to confirm the novelty of this new line. Also: No mention of the users, as well as examples of successful experiments. So the question comes, has the setup been tested ? Is it on the design phase ? It is operational ? These points should be addressed in the discussion, also clearly stating the *novelty* of this new setup with other existing setups in the world.

a graph was added with a spectrum. also the paper states the demagnification, vibration data, the SE suppression, a paper with high throughput results [4]... so the details are tested and evaluated.

Reviewer 3 Report

Comments to authors:

This manuscript reports the design, building and tuning of a new microbeam line dedicated to uIBA experiments. Authors describe the different parts of the microbeam line and why/how they adopted selected technical solutions. Authors were extremely meticulous and convincing for the mechanical design and alignment procedures, but are less conclusive for the IBA tests.

The manuscript suffers from some minor language problems and typo errors that I have listed below together with some scientific queries and comments.

I recommend a minor revision.

Line13 : « Tolerances limit the device accuracy to 3% for RBS” sentence is not clear, see comment regarding line 346.

Line 17: “in-operando” à operando

Line 19-20: “Special apertures and energy resolutions down to 11 keV enable separation of Fe and Cr in RBS.” not clear. see comment regarding line 419.

Line 49: a dot is missing at the end of the sentence.

Lines 55-61: “Unfortunately, the available accelerator infrastructure limited the achievable spot-sizes.”. it’s a pitty that -scarce- information regarding the accelerator is obtained only at the end of the manuscript (line 431). Please give more details.

Line 73: “which will be elaborated, in the following text.”à which will be described in the following paragraphs”.

Line 73: I’m wondering why the setup is called “μNRA” since all other techniques are available.

Line 75: improve the figure quality

Line 82-83: Fig 2a, provide a photography instead of the sketch.

Line 88: “The beamline made from CF100” à “The beamline is made from CF100”

Line 91: “Two faraday cups” à “Two Faraday cups”

Line 93: “first aperture ». Although I suppose this aperture correspond to the object for the focusing lens, it should be clearly specified.

Line 95: “Both apertures feature a fixed 5 mm diameter hole followed by four motor controlled blocks.” I suppose that motors drive apertures, please provide more details.

Line 117: “nuclear reaction products ». Do you mean charged particles? If yes, you should include also scattered particles. However, a residual mT field should not affect reaction product trajectories towards detector apertures; it should be in the um range.

Line 144: “is considered the relevant quantity” à “is considered as the relevant quantity”

Line 143-152: I’m wondering why in your calculation you substract the expansion coefficient of Aluminum and stainless steel. Does Aluminum refers to the supporting table?

Line 164-165 : « The beam-line towards the accelerator employs an identical vacuum system”. This information was already given line 90. Avoid repetition.

Line 177: “The Laser is located” à “The laser is located”

Lines 206-225 This paragraph is not extremely clear. An explanatory sketch could help (in particular to understand the effective position of the mirror relative to sample holder). One of the question is if the optical setup allows beam scintillation monitoring without sample tilting. If there is no room for this supplementary sketch, Fig. 6 could be removed since it is a standard and well-known arrangement.

The next paragraph related to imaging calibration and resolution measurements is also a bit confusing, since authors evoke spatial and optical resolutions (with significant differences: 5.71 and 19.7 um). Is spatial resolution the one achieved on a test stand and the optical resolution the one in the uNRA chamber? Please clarify.

Line 228: “The best spatial resolution was found by varying the parameters angle…”

Line 295: “LiAlO2 » 2 as subscript.

Lines 297-309: in the described procedure, you associate the beam size estimation to scintillation light fading. Do you adjust apertures during the focusing procedure? If yes, light fading can also originates from current falling. Please give more details.

Line 346: “deviations of the measurement results in the order of 3%.” These 3% are related to the one announced in the abstract. You should precise if this deviation correspond to scattering angle uncertainties, it does not necessarily mean the same level of error in the IBA analysis.

Paragraph lines 350-368: this part is clearly not convincing and need a deep revision.

Line 352-353: “two silicon detectors are installed at a reaction angle of 150±2° for separating different particle species” this sentence may be misunderstood in the sense that the use of two detectors allows particle species separation. Please consider rewording.

Line 356-357 : 3.5 MeV protons and 15 MeV 4He ranges in silicon are 120 and 134 um respectively. Please revise the numbers. You claim that its effective thickness is estimated to 300 um since you can detect protons up to 6 MeV. That’s a huge difference! How can you explain this? Moreover, even if the detector active thickness is only 100 um, you should still detect protons of 6 MeV (but with partial energy deposition). Please clarify this part.

Line 364-368: the claimed accuracy regarding NRA detector solid angle seems too optimistic (1%) and there’s no indication of error source. The 7Li(p,α)4He published cross sections are given with typically 3-10% precision and exhibit significant discrepancies between authors. A lack of knowledge about the thin film used here does not help at understanding the procedure (e.g. to what level of precision the Li contents of the film is known), and reaction statistics should be relatively moderate. Finally, the alignment uncertainties and charge measurement errors will certainly contribute to a significant larger solid angle uncertainty.

Line 385: “This value consists of the detector solid angle and the incident ion charge/dose” à “This value is the product of the incident ion charge/dose and detector solid angle.”

Line 402: “Table 2. Measured sample current for two different ion beam setting” à “Table 2. Measured sample current for two different ion beam settings”

Line 405: Following paragraphs sound more as conclusion rather than discussion. Please change the section name.

Line 419: “The good energy resolution in combination with the low geometrical straggling enable a surface near separation of Fe and Cr in steels using RBS.” This sentence is not supported by experimental evidences and miss details about the experimental parameters. It should be removed (also in the abstract). Unless you provide the associated spectrum and missing details.

Ref. section:

Of 8 references, 4 are self-citations. And since a microbeam line is not a novelty by itself, you should cite some of the numerous uIBA setups around the world.

Author Response

Dear Reviewer 3,

thanks for your comments. All suggestions were implemented, see responses below:

Line13 : « Tolerances limit the device accuracy to 3% for RBS” sentence is not clear, see comment regarding line 346.

see reply to line 346

Line 17: “in-operando” à operando

ok

Line 19-20: “Special apertures and energy resolutions down to 11 keV enable separation of Fe and Cr in RBS.” not clear. see comment regarding line 419.

graph was added

Line 49: a dot is missing at the end of the sentence.

added

Lines 55-61: “Unfortunately, the available accelerator infrastructure limited the achievable spot-sizes.”. it’s a pitty that -scarce- information regarding the accelerator is obtained only at the end of the manuscript (line 431). Please give more details.

details added

Line 73: “which will be elaborated, in the following text.”à which will be described in the following paragraphs”.

ok

Line 73: I’m wondering why the setup is called “μNRA” since all other techniques are available.

I better use one term to describe what I do to the people handing in samples than four.

Line 75: improve the figure quality

this must be either your settings or the MDPI system. the original figure is a vector graphic

Line 82-83: Fig 2a, provide a photography instead of the sketch.

I believe the CAD drawing is easier to understand because it has less unnecessary details

Line 88: “The beamline made from CF100” à “The beamline is made from CF100”

changed

Line 91: “Two faraday cups” à “Two Faraday cups”

changed

Line 93: “first aperture ». Although I suppose this aperture correspond to the object for the focusing lens, it should be clearly specified.

added

Line 95: “Both apertures feature a fixed 5 mm diameter hole followed by four motor controlled blocks.” I suppose that motors drive apertures, please provide more details.

more details were added

Line 117: “nuclear reaction products ». Do you mean charged particles? If yes, you should include also scattered particles. However, a residual mT field should not affect reaction product trajectories towards detector apertures; it should be in the um range.

Yes charged particles. Text was amended. We saw a relevant impact on the beam direction and then decided to eliminate the problem once and for all.

Line 144: “is considered the relevant quantity” à “is considered as the relevant quantity”

corrected

Line 143-152: I’m wondering why in your calculation you substract the expansion coefficient of Aluminum and stainless steel. Does Aluminum refers to the supporting table?

Yes, different materials are used here (Bosch X-profiles vs. vacuum parts)

Line 164-165 : « The beam-line towards the accelerator employs an identical vacuum system”. This information was already given line 90. Avoid repetition.

removed

Line 177: “The Laser is located” à “The laser is located”

corrected

Lines 206-225 This paragraph is not extremely clear. An explanatory sketch could help (in particular to understand the effective position of the mirror relative to sample holder). One of the question is if the optical setup allows beam scintillation monitoring without sample tilting. If there is no room for this supplementary sketch, Fig. 6 could be removed since it is a standard and well-known arrangement.

A reference to fig.2 was added and the text slightly rearranged

The next paragraph related to imaging calibration and resolution measurements is also a bit confusing, since authors evoke spatial and optical resolutions (with significant differences: 5.71 and 19.7 um). Is spatial resolution the one achieved on a test stand and the optical resolution the one in the uNRA chamber? Please clarify.

its spatial calibration (µm/pixel) and spatial resolution (µm), wording changed

Line 228: “The best spatial resolution was found by varying the parameters angle…”

corrected

Line 295: “LiAlO2 » 2 as subscript.

corrected

Lines 297-309: in the described procedure, you associate the beam size estimation to scintillation light fading. Do you adjust apertures during the focusing procedure? If yes, light fading can also originates from current falling. Please give more details.

No in this case only the magnets are tuned. Its a stepwise process

Line 346: “deviations of the measurement results in the order of 3%.” These 3% are related to the one announced in the abstract. You should precise if this deviation correspond to scattering angle uncertainties, it does not necessarily mean the same level of error in the IBA analysis.

True, the 3% relate to the Rutherford cross-section. Indeed deviations of other cross-sections will be usually smaller and a general statement is impossible. I see RBS as a gold-standard, though, since the known and high cross-sections usually results in the smallest uncertainties compared to other IBA methods. When combining several results the total uncertainty will mathematically always be larger than the largest individual uncertainty, giving a certain justification to the 3% statement/generalisation. A few sentences were added.

Paragraph lines 350-368: this part is clearly not convincing and need a deep revision.

I understood this comment as a frame around the following comments.

Line 352-353: “two silicon detectors are installed at a reaction angle of 150±2° for separating different particle species” this sentence may be misunderstood in the sense that the use of two detectors allows particle species separation. Please consider rewording.

Yes this is actually the idea. In particular the foil on one detector introduces a dE/dx filter which allows for species separation with two energy resolving detectors.

Line 356-357 : 3.5 MeV protons and 15 MeV 4He ranges in silicon are 120 and 134 um respectively. Please revise the numbers. You claim that its effective thickness is estimated to 300 um since you can detect protons up to 6 MeV. That’s a huge difference! How can you explain this? Moreover, even if the detector active thickness is only 100 um, you should still detect protons of 6 MeV (but with partial energy deposition). Please clarify this part.

I cant explain the thickness mismatch, maybe the supplier just sent me what they have. "Full energy" explanation added

Line 364-368: the claimed accuracy regarding NRA detector solid angle seems too optimistic (1%) and there’s no indication of error source. The 7Li(p,α)4He published cross sections are given with typically 3-10% precision and exhibit significant discrepancies between authors. A lack of knowledge about the thin film used here does not help at understanding the procedure (e.g. to what level of precision the Li contents of the film is known), and reaction statistics should be relatively moderate. Finally, the alignment uncertainties and charge measurement errors will certainly contribute to a significant larger solid angle uncertainty.

cross-section errors drop out, since both detectors measure at the same angle the same ~8MeV alpha products. With counting statistics of 10000 we achieve 1% accuracy. This sentence only speaks about the relative calibration of both detectors, since one of them is equipped with a foil preventing RBS detection. Absolute solid angle calibration is then only required for one detector and can be conducted using RBS.

Line 385: “This value consists of the detector solid angle and the incident ion charge/dose” à “This value is the product of the incident ion charge/dose and detector solid angle.”

changed

Line 402: “Table 2. Measured sample current for two different ion beam setting” à “Table 2. Measured sample current for two different ion beam settings”

corrected

Line 405: Following paragraphs sound more as conclusion rather than discussion. Please change the section name.

true

Line 419: “The good energy resolution in combination with the low geometrical straggling enable a surface near separation of Fe and Cr in steels using RBS.” This sentence is not supported by experimental evidences and miss details about the experimental parameters. It should be removed (also in the abstract). Unless you provide the associated spectrum and missing details.

A spectrum was added with experimental details.

Ref. section:

Of 8 references, 4 are self-citations. And since a microbeam line is not a novelty by itself, you should cite some of the numerous uIBA setups around the world.

True its not a novelty. Another citation added.

Round 2

Reviewer 2 Report

The manuscript has improved, however there are still things that need to be addressed. I recommend publication after minor revision, especially towards clarity of using a well-established terminology instead of terms that are only understood inside the low-energy ion beam community.

Author Response

1
General :
I read with interest the revised version of the manuscript by S. Moller et al., “A new
high-throughput focused MeV ion-beam analysis setup”. The quality with the
manuscript has improved, and I am satisfied with the way that the authors replied to
most (but not all!) of my comments, I thank them for the explanations and
congratulate them again for the research and work done with little resources. At the
same time, there are still a few points that need to be clarified / amended before
publication. I recommend a minor revision of all the points below.
Line 28: I understand from the authors’ reply “kinetic energy without
normalization” means “kinetic energy per ion”. Please add the “per ion” at the end,
because especially for different projectiles of different A and Z, the kinetic energy
per charge or per nucleon can be quite different.
Line 33: Put the “microbeam variants” inside “ “

Ok
Line 44: ...May yield the same spectra may result to the same spectrum.

changed
Line 54: Add a “the” before the ion-beam and end-station properties.

added

Line 57: Sorry, but still the way that this is written is extremely confusing for the
broader audience that “Instruments” has. The phrase is general, it does not matter if
the accelerator is simple or complex. The message to the reader here is that “using
appropriate focusing (quadrupoles) you control the beam “properties” (or the
“betatron oscillations”) based on the beam that you get from the machine
(“emittance”). The word “contradictions” is not a good choice (what are the
contradictions? The small spot-size and the high-brightness? They are not
contradictory requirements – contradictory would be a parallel beam with small
spot-size, e.g). Unless I really missed the message by the authors here, this needs to
be rephrased. E.G : “This can be achieved with proper beam optics that take into
account the initial beam emittance from the accelerator and flexibly focus the beam
on the samples”.
Here again: The brightness of the beam is a function of the spot-size : B ~ d^I/dx’dA,
where dx’ is the beam divergence and dx is the surface of the spot. Maybe I
misunderstood the authors’ reply : They mean that BECAUSE the intensity of their
beam was lower, despite the fact that the demagnification is larger than [7], the
overall brightness is smaller ? If this is the case, then please rephrase :
2
“....1um by combining correct focusing, which, combined with modern ion sources
can lead to extremely high-brightnesses.

Now its clearer to me what you meant. The contradiction originally meant is that an unfocused beam can be collimated for smaller spot sizes but then you lose beam current which is the relevant quantity for our analysis. Many setups for ion beam analysis do not have beam optics.

the paragraph was changed. Hope its clear now.

Line 65: Thanks for adding, now it is clear and completes my previous comment.
Here, again, please use the correct terminology : “limited the spot-sizes” is not
appropriate, because it is not clear why. Please rephrase :
“In our facility, the ions produced by the 1.7 MV CW tandem aceclerator with a
duoplasmatic source (dating in 1983) have usually large spatial and angular profiles (of the order of ~xxx cm and ~x mrad) (“emittances”), a factor that limits the
brightness that can be achieved with the new device. If you don’t know the number,
please put there your estimation.

added

Line 69: A triple quadrupole focusing magnet is not correct. Please change to “a
quadrupole triplet”.

changed

Line 76: “conflicts” Challenges.

I see your point, but i dont like the term challenge here, because it cannot be solved. You can only move on a line between e.g. optimal resolution or count rate. Ok you can find a design point which fits to other limitations, but the singular problem cannot be solved with the given technology.

Figure 1: Much clearer now ! The box under the “beam line” is not clear. Is it a
bulding ? is ta vacuum ?

Its also a vacuum system as the other identical box.

Line 100: X0 corresponds to the amount of material that is seen by a beam, it is
defined as X(the thickness of the material) vs X0(the radiation length, a property of
the material). No problem since they can be moved out of the beam, then forget my
comment. Add a “the” between “during” and “sample analysis”.

added. I didnt know the X0 term. Dont you use SRIM to calculate the range?

Line 102: Change “supplied by” to “powered by”.

changed

Line 103: I missed this during the first review, sorry : Change “resolution” to
“tolerance” for the power supplies.

its the digital steps. changed

Lines 105-106: OK! Now it is clear. These are collimators all together. Please
rephrase towards clarity “Together, these apertures are collimating the beam in
terms of spot-size and divergence”.

changed

Line 108: Add “for limiting the radiation levels to < 10 uSv/h ..... Remove the
duplicate “in particular”. Why you mention here the H, D. And He ions ? In the
other cases you have lower doses ? Please explain.
3

changed. added an explanation

Line 115: type K type “K”. Add “self-produced” before “thermocouples. “

added

Line 148: Between the accelerator, apertures and sample.

added

Figure 5b : Reduce the size of the ticks so that in the x-axis are more nice – maybe
make them bold ?

halfed the number of ticks and increased the font size by 1

Line 162: focus lengthfocal length

changed

Line 292: peak tip fieldThe field at the pole tip is the maximum. So maybe
write “maximum field at the pole tip equal to 0.4T” (remove up-to). The maximum
gradient therefore is (0.4/half/aperture). Thanks for the explanation.

ok

Line 294: Add a “the” Between THE last magnet and target by the
manufacturer of the magnets ? I don’t get this phrase at all. Maybe a typo ?

added

Line 301: “Impact angle spread”. Do the authors mean the “angular spread of the
impinging ions” ? If yes, please rephrase ; “Line focus” is not a clear term. You
mean for an “elongated beam spot” contrary with “circular” beam spot ?

rephrased

Line 304: Phrase not clear. Which detectors width ? They do not appear in Figure
6. Do the authors mean that the detectors’ aperture is much smaller anyway,
therefore the maximum 1.2 degrees will be never reached ?

its about energy resolution of the products. rephrased

Line 308: This phrase should be put up. What does it mean “horizontal beam spot”
? A beam spot has a horizontal dimension and a vertical one. Do the authors mean
“horizontally elongated”, “vertically elongated” or “symmetric” ? If yes, please
rephrase.

rephrased

Line 357: Mentioned also in my previous comments. You have to explain what is
“weight acceptance”, in the sense: “The sample weight acceptances is 300 to 600
g, limited by the capibilties of the motor”.

added
Line 436: The units of Particle *Sr is p/cm2 * sr ? Please clearly mention the
unity.

its a special unit originating from the detection rate of nuclear products being proportional to the incident ions and the detector extent. The unit is particles*Sr (no area). phrase added
Line 445: The authors need do add why they bring diamond as an example. Please
add an explanatory phrase.

its about its electrical resistance with leads to charge accumulation. phrase added